# A Scalable Framework for Table of Contents Extraction from Complex ESG Annual Reports

**Xinyu Wang**[1,2], **Lin Gui**[2], **Yulan He**[1,2,3]

[1]Department of Computer Science, University of Warwick
[2]Department of Informatics, King's College London
[3]The Alan Turing Institute
Xinyu.Wang.11@warwick.ac.uk
{lin.1.gui, yulan.he}@kcl.ac.uk

## Abstract

Table of contents (ToC) extraction centres on structuring documents in a hierarchical manner. In this paper, we propose a new dataset, ESGDoc, comprising 1,093 ESG annual reports from 563 companies spanning from 2001 to 2022. These reports pose significant challenges due to their diverse structures and extensive length. To address these challenges, we propose a new framework for Toc extraction, consisting of three steps: (1) Constructing an initial tree of text blocks based on reading order and font sizes; (2) Modelling each tree node (or text block) independently by considering its contextual information captured in node-centric subtree; (3) Modifying the original tree by taking appropriate action on each tree node (*Keep*, *Delete*, or *Move*). This construction-modelling-modification (CMM) process offers several benefits. It eliminates the need for pairwise modelling of section headings as in previous approaches, making document segmentation practically feasible. By incorporating structured information, each section heading can leverage both local and long-distance context relevant to itself. Experimental results show that our approach outperforms the previous state-of-the-art baseline with a fraction of running time. Our framework proves its scalability by effectively handling documents of any length.[1]

## 1 Introduction

A considerable amount of research has been proposed to comprehend documents (Xu et al., 2019; Zhang et al., 2021; Xu et al., 2021a,b; Peng et al., 2022; Li et al., 2022; Gu et al., 2022; Shen et al., 2022; Lee et al., 2022, 2023) , which typically involves the classification of different parts of a document such as title, caption, table, footer, and so on. However, such prevailing classification often centres on a document's local layout structure, sidelining a holistic comprehension of its content

and organisation. While traditional summarisation offers a concise representation of a document's content, a Table of Contents (ToC) presents a structured and hierarchical summary. This structural organisation in a ToC provides a comprehensive pathway for pinpointing specific information. For example, when seeking information about a company's carbon dioxide emissions, a ToC enables a systematic navigation through the information hierarchy. In contrast, conventional summarisation might only provide a vague indication of such information, requiring sifting through the entire document for precise detail.

Several datasets have been proposed to facilitate the research in document understanding (Zhong et al., 2019b; Li et al., 2020; Pfitzmann et al., 2022). Most of these studies lack a structured construction of documents and primarily focus on well-structured scientific papers. A dataset called Hier-Doc (Hierarchical academic Document) (Hu et al., 2022) was introduced to facilitate the development of methods for extracting the table of contents (ToC) from documents. This dataset was compiled from scientific papers downloaded from arXiv[2], which are typically short and well-structured. The hierarchical structure can often be inferred directly from the headings themselves. For example, the heading "*1. Introduction*" can be easily identified as a first-level heading based on the section numbering. Moreover, due to the relatively short length of scientific papers, it is feasible to process the entire document as a whole. Hu et al. (2022) proposed the multimodal tree decoder (MTD) for ToC extraction from HierDoc. MTD first utilises text, visual, and layout information to encode text blocks identified by a PDF parser; then classifies all text blocks into two categories, headings and non-headings; and finally predicts the relationship of each pair of headings, facilitating the parsing of these headings into a tree structure representing ToC.

---

[1]Available at https://github.com/xnyuwg/cmm.

[2]https://arxiv.org/

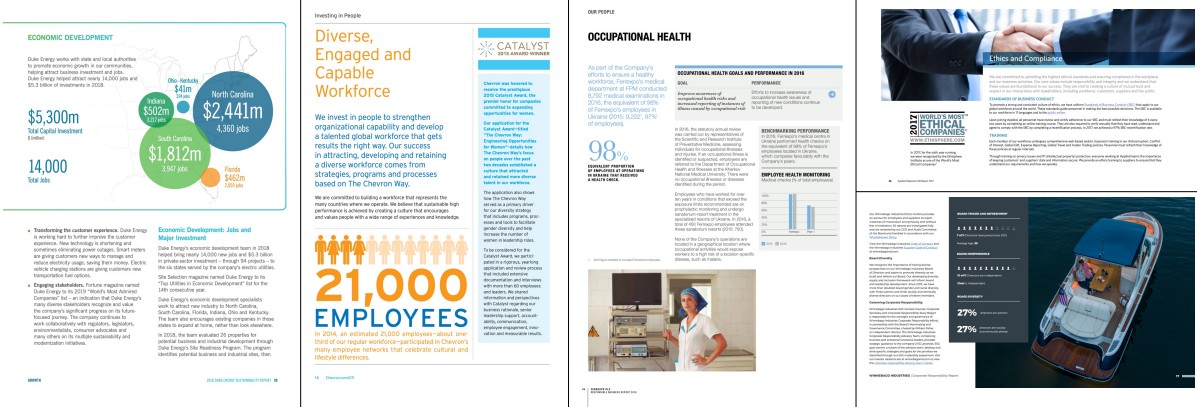

Figure 1: Five examples of ESG reports, with the left three presented in portrait orientation and the right-most two in landscape orientation. They show a wide range of diverse structures. It is common to observe the absence of section numbering.

However, understanding long documents such as ESG (Environmental, Social, and Governance) annual reports poses significant challenges compared to commonly used scientific papers. First, ESG reports tend to be extensive, often exceeding 100 pages, which is uncommon for scientific papers. Second, while scientific papers generally adhere to a standard structure that includes abstract, introduction, methods, results, discussion, and conclusion sections, ESG reports exhibit more diverse structures with a wide range of font types and sizes. Third, ESG reports often include visual elements such as charts, graphs, tables, and infographics to present data and key findings in a visually appealing manner, which adds complexity to the document parsing process. Some example ESG reports are illustrated in Figure 1.

In this paper, we develop a new dataset, ESGDoc, collected from public ESG annual reports[3] from 563 companies spanning from 2001 to 2022 for the task of ToC extraction. The existing approach, MTD (Hu et al., 2022), faces difficulties when dealing with challenges presented in ESGDoc. MTD models relationships of every possible heading pairs and thus requires the processing of the entire document simultaneously, making it impractical for lengthy documents. As will be discussed in our experiments section, MTD run into out-of-memory issue when processing some lengthy documents in ESGDoc. Moreover, MTD only uses Gated Recurrent Unit (GRU) (Cho et al., 2014) to capture the context of a section heading, lacking long-distance interaction, particularly for high-level headings that may be tens of pages apart.

In order to overcome the challenges presented in ESGDoc, we propose a new scalable framework, consisting of three main steps: (1) Constructing an initial tree of text blocks based on reading order and font sizes; (2) Modelling each tree node (or text block) independently by considering its contextual information captured in node-centric subtree; (3) Modifying the original tree by taking appropriate action on each tree node (*Keep*, *Delete*, or *Move*). Our method is named as CMM (Construction-Modelling-Modification).

This approach allows higher-level headings to focus on capturing high-level and long-distance information, while lower-level headings focus more on local information. Additionally, CMM also models each heading independently, removing the need for modelling pairwise relationships among headings and enabling more effective document segmentation. Here, we can divide documents based on the tree structure instead of relying on page divisions. This ensures that each segment maintains both local and long-distance relationships, preserving the long-distance connections that would be lost if division were based on page boundaries. As CMM does not require the processing of a document as a whole, it can be easily scaled to deal with lengthy documents. Experimental results show that our approach outperforms the previous state-of-the-art baseline with only a fraction of running time, verifying the scalability of our model as it is applicable to documents of any length. Our main contributions are summarised as follows:

- We introduce a new dataset, ESGDoc, comprising 1,093 ESG annual reports specifically designed for table of contents extraction.

- We propose a novel framework that processes documents in a construction-modelling-modification manner, allowing for the decoupling of each heading, preserving both local and long-distance relationships, and incorporating structured information.
- We present a novel graph-based method for document segmentation and modelling, enabling the retention of both local and long-distance information within each segment.

## 2 Related Work

**Datasets** Many datasets have been proposed for document understanding. PubLayNet dataset (Zhong et al., 2019b) is a large-scale dataset collected from PubMed Central Open Access, which uses scientific papers in PDF and XML versions for automatic document layout identification and annotation. Article-regions dataset (Soto and Yoo, 2019) offers more consistent and tight-fitting annotations. DocBank dataset (Li et al., 2020) leverages the latex source files and font colour to automatically annotate a vast number of scientific papers from arXiv. DocLayNet dataset (Pfitzmann et al., 2022) extends the scope from scientific papers to other types of documents. However, these datasets primarily contain annotations of the type and bounding box of each text, such as title, caption, table, and figure, but lack structured information of documents.

**Approaches for Document Understanding** In terms of methods for document understanding, a common approach is the fusion of text, visual, and layout features (Xu et al., 2019; Zhang et al., 2021; Xu et al., 2021a,b; Peng et al., 2022; Li et al., 2022), where visual features represent images of texts and the document, and layout features comprise bounding box positions of texts. Some methods also introduced additional features. For instance, XY-LayoutLM (Gu et al., 2022) incorporates the reading order, VILA (Shen et al., 2022) utilises visual layout group, FormNet (Lee et al., 2022, 2023) employs graph learning and contrastive learning. The aforementioned methods focus on classifying individual parts of the document rather than understanding the structure of the entire document.

**Table of Contents (ToC) Extraction** In addition to document understanding, some work has been conducted on the extraction of ToC. Early methods primarily relied on manually designed rules to extract the structure of documents (Nambood-

iri and Jain, 2007; Doucet et al., 2011). Tuarob et al. (2015) designed some features and use Random Forest (Breiman, 2001) and Support Vector Machine (Bishop and Nasrabadi, 2006) to predict section headings. Mysore Gopinath et al. (2018) propose a system for section titles separation. MTD (Hu et al., 2022) represents a more recent approach, fusing text, visual, and layout information to detect section headings from scientific papers in the HierDoc (Hu et al., 2022) dataset. It also uses GRU (Cho et al., 2014) and attention mechanism to classify the relationships between headings, generating the tree of ToC. While MTD performs well on HierDoc, it requires modelling all headings in the entire document simultaneously, which is impractical for long documents. To address this limitation, we propose a new framework that decouples the relationships of headings for ToC extraction and introduces more structural information by utilising font size and reading order, offering a more practical solution for long documents.

## 3 Dataset Construction

To tackle the more challenging task of ToC extraction from complex ESG reports, we construct a new dataset, ESGDoc, from ResponsibilityReports.com[4]. Initially, we have downloaded 10,639 reports in the PDF format. However, only less than 2,000 reports have ToC in their original reports. To facilitate the development of an automated method for ToC extraction from ESG reports, we selectively retrain reports that already possess a ToC. The existing ToC serves as the reference label for each ESG report, while the report with the ToC removed is used for training our framework specifically designed for ToC extraction.

Our final dataset comprises 1,093 publicly available ESG annual reports, sourced from 563 distinct companies, and spans the period from 2001 to 2022. The reports vary in length, ranging from 4 pages to 521 pages, with an average of 72 pages. In contrast, HierDoc (Hu et al., 2022) has a total of 650 scientific papers, which have an average of 19 pages in length. We randomly partitioned the dataset into a training set with 765 reports, a development set with 110 reports, and a test set with 218 reports.

Text content from ESG reports was extracted using PyMuPDF[5] in a format referred to as "block". A block, defined as a text object in the PDF stan-

---

[4] https://www.responsibilityreports.com/
[5] https://pymupdf.readthedocs.io/

dard, which is usually a paragraph, can encompass multiple lines of text. We assume that the text within a text object is coherent and should be interpreted as a cohesive unit. Each block comprises the following elements: *text content*, *font*, *size*, *colour*, *position*, and *id*. The *id* is a unique identifier assigned to each block to distinguish blocks that contain identical *text content*. The *position* refers to the position of the block within a page in the ESG PDF report, represented by four coordinates that denote the top-left and bottom-right points of the block bounding box. Other elements, such as *font*, *size*, and *colour*, provide additional information about the text.

## 4 Methodology

We propose a framework for ToC extraction based on the following assumptions:

**Assumption 1** *Humans typically read documents in a left-to-right, top-to-bottom order, and a higher-level heading is read before its corresponding sub-heading and body text.*

**Assumption 2** *In a table of contents, the font size of a higher-level heading is no smaller than that of a lower-level heading or body text.*

**Assumption 3** *In a table of contents, headings of the same hierarchical level share the same font size.*

In our task, a document is defined as a set of blocks. To replicate the reading order of humans, we reorder the blocks from the top-left to the bottom-right of the document. We employ the XY-cut algorithm (Ha et al., 1995) to sort the blocks. The sorted blocks are denoted as $\{x_i\}_{n=1}^{n_b}$, where $\{x_{<i}\}$ precedes $x_i$ and $\{x_{>i}\}$ follows $x_i$. Here, $n_b$ represents the total number of blocks. For each block $x_i$, we define $s_i$ as its *size*.

**Problem Setup**   Given a list of blocks, ToC extraction aims to generate a tree structure representing the table of contents, where each node corresponds to a specific block $x$. We introduce a pseudo root node $r$ as the root of the ToC tree.

We propose to initially construct a full tree containing all the blocks, where the hierarchical relation between blocks is simply determined by their respective font sizes. Specifically, when two blocks, $x_i$ and $x_j$, are read in sequence, if they are close to each other and their font sizes $s_i > s_j$, then $x_j$ becomes a child of $x_i$. We then modify the tree by removing or rearranging nodes as necessary. Essentially, for a node (i.e., a block) $x_i$, we need to

learn a function which determines the operation ('*Keep*', '*Delete*', or '*Move*') to be performed on the node. In order to enable document segmentation and capture the contextual information relating to the node, our approach involves extracting a subtree encompassing its neighbourhood including the parent, children and siblings, within a range of $n_d$ hops. Subsequently, we use Graph Attention Networks (GATs) (Brody et al., 2021; Velickovic et al., 2017) to update the node information within the subtree. An overview of our proposed framework is illustrated in Figure 2.

Before delving into the detail of our proposed framework, we first define some notations relating to node operations. $PA(x_i)$ as the parent node of node $x_i$, $PR(x_i)$ as the preceding sibling node of $x_i$. $SU(x_i)$ as the subsequent sibling node of $x_i$. We also define $PRS(x_i)$ as all the preceding sibling nodes of $x_i$.

### 4.1 Tree Construction

We first construct a complete tree $\mathcal{T}$, consisting of all identified blocks using PyMuPDF, based on reading order and font sizes. For each node $x_i$, we find a node in its previous nodes $x_j \in \{x_{<i}\}$ that is closest to $x_i$ and $s_j > s_i$. Then $x_i$ becomes a child of $x_j$. A detailed algorithm is in Appendix A. Following the principles outlined in Assumptions 1, 2 and 3, this approach assumes that the ToC is contained within the tree structure, as shown in the top-left portion of Figure 2. The subsequent steps of our model involve modifying this tree $\mathcal{T}$ to generate the ToC.

### 4.2 Tree Modelling

In this section, for a given tree node, we aim to learn a function which takes the node representation as input and generates the appropriate operation for the node. In what follows, we first describe how we encode the contextual information of a node, and then present how to learn node representations.

**Node-Centric Subtree Extraction**   To effectively encode the contextual information of a tree node and to avoid processing the whole document in one go, we propose to extract a node-centric subtree, $\boldsymbol{t}_i$, a tree consisting of neighbourhood nodes, including $PR(x_i)$ and $SU(x_i)$, of node $x_i$, extracted via Breadth First Search (BFS) on $x_i$ with a depth $n_d$. Here, $n_d$ is a hyper-parameter. The neighbourhood nodes consist of the parent node,

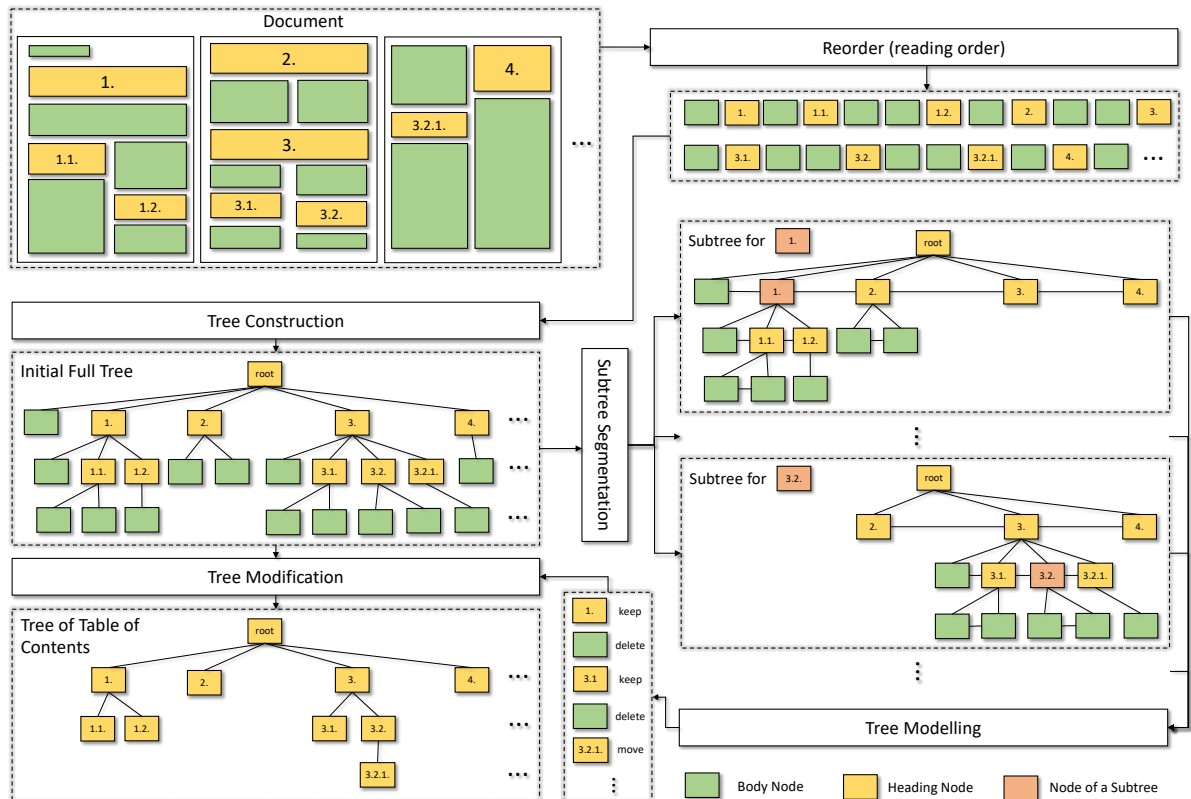

Figure 2: Overview of CMM. Initially, blocks across multiple pages in a document shown in **top-left**, are reordered into a sequence based on reading order (**Top-right**). A full tree consisting of all text blocks is constructed based on reading order and font size (**Center-left**). Subsequently, for each node, a node-centric subtree is extracted and modelled by a graph neural network (GNN). In the subtree shown in **center-right**, the first-level heading '*1.*' can access both long-distance relationships with other first-level headings '*2.*', '*3.*', '*4.*', and local relationships with heading '*1.1*', '*1.2*', and bodies. In contrast, in the subtree shown in **bottom-right**, the second-level heading '*3.2.*' concentrates more on local information but also has access to some global information. When modelling with GNN, each node is connected with its neighbourhood nodes, including parent, children and siblings. After the tree modelling phase, the node-level operations (*Keep*, *Delete*, and *Move*) are predicted. The original tree is then modified according to the node-level operations, resulting in the final Table of Contents (**Bottom-left**). The nodes numbered and coloured in this figure are for illustrative purposes; some body nodes are omitted for brevity. During inference, the model is unaware of whether a node is a heading or non-heading.

children nodes, and the sibling nodes of node $x_i$, as shown in the right portion of Figure 2. Apart from the edges linking parent and child, we have additionally added edges connecting neighbouring sibling nodes.

**Node Encoding**    Before discussing how to update node representations in a subtree, we first encode each node (or block) $x_i$ into a vector representation. We employ a text encoder, which is a pre-trained language model, to encode the *text content* of $x_i$. We also utilise additional features from $x_i$ defined as $f_i$. These features include: (1) pdf page number; (2) font and font size; (3) colour as RGB; (4) the number of text lines and length; and (6) the position of the bounding box of the block, represented by the coordinates of the top-left and bottom-right

points. The representation of $x_i$ is derived from text encoder and $f_i$ with a Multilayer Perceptron (MLP) as follows:

$$b_i = \text{MLP}([\text{TextEncoder}(x_i), f_i]) \qquad (1)$$

where $b_i$ denotes the hidden representation of block $x_i$ and $[.,.]$ denotes concatenation.

To simulate the human reading order, we apply a one-layer bi-directional Gated Recurrent Unit (GRU) (Cho et al., 2014) on nodes of the subtree with in-order traversal as follows:

$$\{v_i\}^{|t_i|} = \text{GRU}(t_i) \qquad (2)$$

where $\{v_i\}^{|t_i|}$ denotes all hidden representations of the nodes in $t_i$ after GRU encoding.

**Node Representation Update in a Subtree**   We transform each node-centric subtree $t_i$ to a graph $\mathcal{G} = (\mathcal{V}, \mathcal{E})$, where the nodes $\mathcal{V} = \{x_j \in t_i\}$, and the embedding of each node $x_j$ is assigned as $v_j$. For each node $x_j$, there are three types of edges, from its parent, from its children, and from its siblings. The edges between the parent/siblings and $x_i$ may span across multiple pages, as headings can be widely separated. Such edges can provide long-distance relationship information. On the other hand, the edges from children to $x_i$ provide localised information about the heading. Thus, $x_i$ benefits from learning from both long-distance and local relationships.

We employ Graph Attention Networks (GAT) with $n_d$ layers for graph learning, enabling each $x_i$ to focus on other nodes that are more relevant to itself. GAT also uses edge embeddings. In our model, we define edge features $f_{j,i}$ for edge $e_{j,i}$ as follows: (1) the edge type (*parent*, *children*, or *siblings*); (2) size difference $s_j - s_i$; (3) whether $x_j$ and $x_i$ have the same font or colour; (4) page difference; (5) position difference as the differences of the coordinates of the top-left and bottom-right points of the corresponding block bounding boxes.

With nodes $\mathcal{V} = \{x_i \in t_i\}$, node embeddings $\{v_i\}$, edge $\mathcal{E} = \{e_{j,i}\}$, and edge embeddings $f_{j,i}$, the graph learning is performed as follows:

$$\{h_i\}^{|t_i|} = \text{GAT}(\mathcal{V}, \mathcal{E}) \qquad (3)$$

where $\{h_i\}^{|t_i|}$ are the hidden representations of nodes $\{x_i\}^{|t_i|}$ in the node-centric subtree $t_i$. In practice, multiple node-centric subtrees can be merged and represented simultaneously in GPU to accelerate training and inference.

### 4.3   Tree Modification

In this section, we discuss how the model predicts and executes modifications to the tree. We define three types of operations for each node:

1. *Delete*: This node is predicted as not a heading and will be deleted from the tree.
2. *Move*: This node is predicted as a low-level heading that is a sibling of a high-level heading due to having the same font size in rare cases. The node *3.2.1* in Figure 2 is an example. This node will be relocated to be a child as its preceding sibling as non-heading nodes have already been deleted.
3. *Keep*: This node is predicted as a heading and does not require any operations.

We define three scores $o_i^{[\text{kp}]}$, $o_i^{[\text{de}]}$ and $o_i^{[\text{mv}]}$ to represent the likelihood that the node $x_i$ should be kept, deleted or moved. These scores are computed as follows:

$$o_i^{[\text{kp}]} = W_{\text{kp}} h_i + b_{\text{kp}}$$
$$o_i^{[\text{de}]} = W_{\text{de}} h_i + b_{\text{de}} \qquad (4)$$
$$o_i^{[\text{mv}]} = W_{\text{mv}}[\text{POOL}(PRS(h_i)), h_i] + b_{\text{mv}}$$

where $\text{POOL}(PRS(h_i))$ denotes a max pooling layer on the representations of preceding siblings of $x_i$; $[.,.]$ denotes the concatenation; $W_{\text{kp}}$, $W_{\text{de}}$, $W_{\text{mv}}$, $b_{\text{kp}}$, $b_{\text{de}}$, and $b_{\text{mv}}$ are learnable parameters. The score of *Keep* and *Delete* is inferred from the node directly, as $h_i$ has gathered neighbourhood information with both long-distance and local relationships. The score of *Move* is inferred from the node and its preceding siblings so that the node can compare itself with its preceding siblings to decide whether it is a sub-heading of preceding siblings.

The probabilities of the node operations are computed with the softmax function as follows:

$$p_i^{[.]} = \frac{e^{o_i^{[.]}}}{e^{o_i^{[\text{kp}]}} + e^{o_i^{[\text{de}]}} + e^{o_i^{[\text{mv}]}}} \qquad (5)$$

where $[.]$ could be $[\text{kp}]$, $[\text{de}]$ or $[\text{mv}]$. The final operation for node $x_i$ is determined as follows:

$$\hat{y}_i = \text{argmax}(\boldsymbol{p}_i) \qquad (6)$$

where each $\hat{y}_i$ could be *Keep*, *Delete*, or *Move*.

For each node-centric subtree $t_i$, the model only predicts the operation $\hat{y}_i$ for node $x_i$ and ignores other nodes. With all $\{\hat{y}_i\}_{i=1}^{n_b}$ predicted for nodes $\{x_i\}_{i=1}^{n_b}$, the original tree $\mathcal{T}$ will be modified as shown in Algorithm 1, where node deletion is performed first, followed by node relocation.

We assume that all non-heading nodes have already been deleted during the deletion step. Each node is then checked following the reading order whether it should be moved. Therefore, for a node to be moved, we can simply set its preceding sibling as its parent node.

The modified tree $\mathcal{T}'$ represents the final inference output of our method, which is a ToC.

### 4.4   Inference and Training

For training the model, we define the ground truth label $y_i$ of operation for each node $x_i$. If a node $x_i$ is not a heading, then its label is $y_i = $ *Delete*. If a node $x_i$ is a heading, and there is a higher-level

**Algorithm 1:** Tree Modification

**Input:** A tree $\mathcal{T}$, node operations $\{\hat{y}_i\}_{i=1}^{n_b}$
$X^{[de]} \leftarrow \{x_i \in \mathcal{T}, \hat{y}_i = \text{'}delete'\}$
$\mathcal{T}' = \{x_i \in \mathcal{T} \setminus X^{[de]}\}$
Reconstruct the tree $\mathcal{T}'$ with reading order and font size.
**foreach** $x_i \in \mathcal{T}'$ **do**
    **if** $\hat{y}_i = \text{'}Move'$ **then**
        $PA(x_i) \leftarrow PR(x_i)$ /* Set the parent of $x_i$ as its preceding sibling    */
    **end**
**end**
**Output:** The modified tree $\mathcal{T}'$

heading in its preceding nodes, then the label is $y_i = Move$. Otherwise, the label is $y_i = Keep$. The loss is the cross entropy between $\hat{y}_i$ and $y_i$.

## 5 Experiments

### 5.1 Experimental Setup

**Baselines** We use MTD (Hu et al., 2022) as our baseline, which utilises multimodal information from images, text, and layout. The MTD consists of two steps: firstly, classifying and selecting headings from documents with a pre-trained language model, and secondly, modelling via GRU (Cho et al., 2014) and decoding heading relations into a hierarchical tree.

**Dataset** We evaluate CMM on the following ToC extraction datasets: **(1) ESGDoc** dataset consists of 1,093 ESG annual report documents, with 765, 110, and 218 in the train, development, and test sets, respectively. In our experiments, MTD encounters out-of-memory issues when processing some long documents in ESGDoc as it needs to model the entire document as a whole. Therefore, we curated a sub-dataset, denoted as **ESGDoc (Partial)**, which consists of documents from ESGDoc that are less than 50 pages in length. This sub-dataset contains 274, 40, and 78 documents in the train, development, and test sets, respectively. **(2) HierDoc** dataset (Hu et al., 2022) contains 650 scientific papers with 350, 300 in the train, and test sets, respectively. Given that the extracted text from HierDoc does not include font size, we extract font size from PDF directly using PyMuPDF.

**Evaluation Metrics** We evaluate our method in two aspects: heading detection (HD) and the tree

of Toc. HD is evaluated using the F1-score, which measures the effectiveness of our method in identifying headings from the document, which primarily relates to construction and modelling steps, as it does not measure the hierarchical structure of ToC. For Toc, we use tree-edit-distance similarity (TEDS) (Zhong et al., 2019a; Hu et al., 2022), which compares the similarity between two trees based on their sizes and the tree-edit-distance (Pawlik and Augsten, 2016) between them:

$$\text{TEDS}(\mathcal{T}_p, \mathcal{T}_g) = 1 - \frac{\text{TreeEditDist}(\mathcal{T}_p, \mathcal{T}_g)}{\max(|\mathcal{T}_p|, |\mathcal{T}_g|)} \quad (7)$$

For each document, a TEDS is computed between the predicted tree $\mathcal{T}_p$ and the ground-truth tree $\mathcal{T}_g$. The final TEDS is the average of the TEDSs of all documents.

**Implementation Detail** We use RoBERTa-base (Liu et al., 2019) as the text encoder model. We set the BFS depth $n_d = 2$, and the hidden size of $b$, $v$, and $h$ to 128. Our model is trained on a NVIDIA A100 80G GPU using the Adam optimizer (Kingma and Ba, 2015) with a batch size 32. We use a learning rate of $1e$-5 for pretrained parameters, and a learning rate of $1e$-3 for randomly initialised parameters. In some instances, the font size may be automatically adjusted slightly depending on the volume of text to ensure that texts that have varying fonts do not share the same font sizes. Texts with very small sizes are automatically deleted during modification.

### 5.2 Assumption Violation Statistics

| Dataset | A1 | A2 | A3 | Any |
|---------|-----|-----|-----|------|
| HierDoc | 0.0 | 0.5 | 4.1 | 4.6 |
| ESGDoc | 0.8 | 1.7 | 8.7 | 10.8 |

Table 1: The percentage (%) that each assumption is violated. **Any** denotes the percentage that the heading violates at least one assumption.

Our method is based on Assumption 1, 2, and 3. However, these assumptions do not always hold. This section presents statistics on the percentage of headings that violate these assumptions by automatically examining consecutive blocks along the sorted blocks $\{x_i\}_{n=1}^{n_b}$ with their labels. As shown in Table 1, there are 4.6% and 10.8% of headings that contravene these assumptions in HierDoc and ESGDoc, respectively. Our current method is unable

to process these non-compliant headings. Despite these limitations, our method still achieves good performance, as will be detailed in Section 5.3.

## 5.3 Overall Results

Table 2 presents the overall TEDS results on Hier-Doc and ESGDoc. Both models demonstrate good performance on HierDoc with CMM slightly outperforming MTD. However, we observe significant performance drop 1.1on ESGDoc, indicating the challenge of processing complex ESG reports compared to scientific papers. MTD exhibits a notably low TEDS score in ESGDoc (Full) due to the out-of-memory issue it encountered when processing certain lengthy documents. To address this, we exclude documents longer than 50 pages, resulting in MTD achieving a TEDS score of $26.9\%$ on ESGDoc (Partial). Nevertheless, our approach CMM outperforms MTD by a substantial margin. The HD F1-score of our method outperforming MTD by 12.8% on ESGDocPartial also demonstrates the effectiveness of construction and modelling steps. Due to the violation of assumptions as discussed in Section 5.2, the improvement of our model over MTD in TOC is less pronounced compared to HD.

| Model | HierDoc | | ESGDoc F. | | ESGDoc P. | |
|---|---|---|---|---|---|---|
| | HD | ToC | HD | ToC | HD | ToC |
| MTD | 96.1 | 87.2 | 12.7 | 12.8 | 40.4 | 26.9 |
| CMM (Ours) | 97.0 | 88.1 | 55.6 | 33.2 | 53.2 | 30.0 |

Table 2: Heading detection (HD) in F1-score and ToC in TEDS (%) of MTD and CMM on HierDoc, ESGDocFull (F.), and ESGDocPartial (P.).

Our model's performance on ESGDoc (Full) demonstrates its scalability in handling ESG reports with diverse structures and significantly lengthy text. The comparable TEDS scores between CMM and MTD on HierDoc can be potentially attributed to the nature of scientific papers. For instance, headings in scientific papers such as "*5 Experiments*" and "*5.1 Experimental Setup*", provide explicit indication of their hierarchical relationships within the headings themselves. The presence of section numbering such as "5" and "5.1" makes it easier to determine their hierarchical level such as the latter being a sub-heading of the former. Our method introduces hierarchical information via the reading order and font sizes, and learns tree node representations by simultaneously considering long-distance and local relationships. However,

if the hierarchical information is already contained in the headings, our method may not offer many additional hierarchical insights.

## 5.4 Run-Time Comparison

| Model | HierDoc | | ESGDoc Partial | |
|---|---|---|---|---|
| | Time | Ratio | Time | Ratio |
| **Training** | | | | |
| MTD | 2420.6 | 4.6x | 513.5 | 2.1x |
| CMM (Ours) | 525.4 | 1.0x | 241.4 | 1.0x |
| **Inference** | | | | |
| MTD | 16.1 | 4.2x | 2.7 | 1.3x |
| CMM (Ours) | 3.8 | 1.0x | 2.1 | 1.0x |

Table 3: The GPU training and inference time in minutes for MTD and CMM on HierDoc and ESGDoc (Partial).

Table 3 presents the run-time comparison between MTD and CMM on HierDoc and ESGDoc (Partial). MTD consumes 4.6x and 2.1x more time for training and 4.2x and 1.3x more time for inference on HierDoc and ESGDoc, respectively. Different from MTD, CMM does not need to model all possible pairs of headings. Instead, it only predicts whether a node should be deleted or relocated, thereby reducing the computational time.

Compared to MTD, our model exhibits higher efficiency on HierDoc compared to ESGDoc. This could be attributed to the larger number of edges in the graphs constructed from node-centric subtrees in our method for ESGDoc. ESG annual reports often contain numerous small text blocks, such as "$5,300m", "14,000", and "3,947 jobs", as illustrated in the first example of Figure 1. Our method treats these individual texts as separate nodes in both the trees and graphs, leading to a significant increase in the number of edges in ESGDoc compared to HierDoc.

## 5.5 Ablation Study

Table 4 illustrates how different components in CMM contribute to performance:

**w/ page-based division**  CMM divides the document into subtrees based on the tree structure. We substitute the tree-based division with a page-based one. Initially, the document is divided using a window of 6 pages with a 2-page overlap. All other steps remain unchanged, including the modelling of node-centric subtrees. The choice of the

| Model | HierDoc | | ESGDoc | |
|---|---|---|---|---|
| | HD | ToC | HD | ToC |
| CMM | 97.0 | 88.1 | 55.6 | 33.2 |
| w/ page division | 96.7 | 87.8 | 52.3 | 30.1 |
| w/o GRU | 96.8 | 87.7 | 50.8 | 30.5 |
| w/o GNN | 96.4 | 87.4 | 44.0 | 24.9 |

Table 4: Ablation Study of CMM on HierDoc and ESGDoc with heading detection (HD) and ToC results reported in F1-score and TEDS (%), respectively.

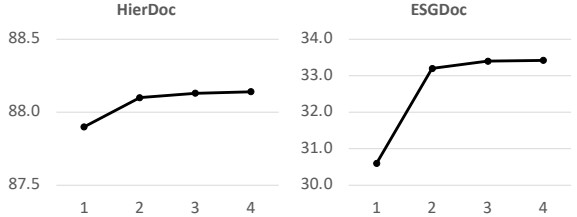

Figure 3: Study of BFS depth $n_d$ of CMM on HierDoc and ESGDoc in TEDS.

page number for division is made to keep a similar GPU memory consumption. This results in a performance drop of $0.3\%$ on HierDoc and $3.1\%$ on ESGDoc. The page-based division impedes long-distance interaction, resulting in a lack of connection between high-level headings.

**w/o GRU** We exclude the GRU in Eq. (2) and directly set $v_i = b_i$. The results show a performance drop of $0.4\%$ on HierDoc and $2.7\%$ on ESGDoc.

**w/o GNN** We exclude the GNN in Eq. (3) and directly set $h_i = v_i$. This results in a more significant performance drop of $0.7\%$ on HierDoc and $8.3\%$ on ESGDoc. With GNN, each heading can gather information from other long-distance and local body nodes effectively and simultaneously.

As shown in Table 4, there is a larger performance drop on ESGDoc compared to HierDoc. This can be attributed to the same reason outlined in Section 5.3: inferring hierarchical relationships from headings themselves is easier in HierDoc than in ESGDoc. Therefore, the removal of components that introduce hierarchical relationships does not significantly harm the performance on HierDoc.

Figure 3 demonstrates how performance varies with different values of $n_d$, the depth of neighbourhood during BFS for constructing node-centric subtrees. Due to the nature of the data, there is limited improvement observed on HierDoc as $n_d$ increases. Notably, there is a substantial increase in TEDS from $n_d = 1$ to $n_d = 2$, but the improvement becomes negligible when $n_d > 2$. Therefore, we select $n_d = 2$ considering the trade-off between performance and efficiency.

The primary factors contributing to the negligible improvement $n_d > 2$ may include: (1) A significant portion of documents exhibit a linear structure. To illustrate, when $n_d = 2$, it corresponds to a hierarchical arrangement featuring primary headings,

secondary headings, and main body content in a three-tier configuration. (2) The constructed initial tree inherently positions related headings in close proximity according to their semantic relationships, without regard for their relative page placement. As a consequence, a heading's most relevant contextual information predominantly emerges from its immediate neighbours within the tree. For example, when examining heading 1.2., information from heading 1. ($n_d = 1$) offers a comprehensive overview of the encompassing chapter. Simultaneously, heading 2. ($n_d = 2$) can provide supplementary insights, such as affirming that heading 1.2. is nested within the domain of heading 1., rather than heading 2. However, delving into deeper levels may become redundant. For example, a heading like 2.2. ($n_d = 3$), situated more distantly in the semantic space, would not notably enhance the understanding of heading 1.1.

Some case studies illustrating the ToC extraction results of CMM on ESGDoc are presented in Appendix C.

## 6 Conclusion and Future Work

In this paper, we have constructed a new dataset, ESGDoc, and proposed a novel framework, CMM, for table of contents extraction. Our pipeline, consisting of tree construction, node-centric subtree modelling, and tree modification stages, effectively addresses the challenges posed by the diverse structures and lengthy nature of documents in ESGDoc. The methodology of representing a document as an initial full tree, and subsequently predicting node operations for tree modification, and further leveraging the tree structure for document segmentation, can provide valuable insights for other document analysis tasks.

## Acknowledgements

This work was funded by the the UK Engineering and Physical Sciences Research Council (grant no. EP/T017112/1, EP/T017112/2, EP/V048597/1). YH is supported by a Turing AI Fellowship funded by the UK Research and Innovation (grant no. EP/V020579/1, EP/V020579/2).

## Limitations

Our method exhibits two primary limitations. Firstly, it relies on the extraction of font size. For documents in photographed or scanned forms, an additional step is required to obtain the font size before applying our method. However, with the prevailing trend of storing documents in electronic formats, this limitation is expected to diminish in significance.

Secondly, our method is grounded in Assumptions 1, 2, and 3. As discussed in Section 5.2, our current method encounters difficulties in scenarios where these assumptions are not met. Some examples of such assumption violations are provided in Appendix B. However, it is worth noting that these assumption violations primarily impact the modification step in our construction-modelling-modification approach. If we focus solely on the construction and modelling steps, our method still outperforms MTD in heading detection. Therefore, future efforts to enhance the modification step, which is susceptible to assumption violations, could hold promise for improving the overall performance of our approach.

## Ethics Statement

The ESG annual reports in `ESGDoc` are independently published by the companies and are publicly accessible. ResponsibilityReports.com[6] compiles these ESG annual reports, which are also accessible directly on the respective companies' websites. There are also other websites such as CSRWIRE[7] and sustainability-reports.com[8] serve as repositories for these reports. Because these reports are publicly available, the use of such data for research purpose is not anticipated to present any ethical concerns.

---

[6] https://www.responsibilityreports.com/
[7] https://www.csrwire.com/reports
[8] https://www.sustainability-reports.com/

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

# Appendix

## A  Algorithm of Building an Initial Tree

Algorithm 2 describes how to build an initial full tree from a document.

---

**Algorithm 2:** Building a Tree based on Reading Order and Font Sizes

---

**Input:** Root node $r$, all blocks $\{x_i\}_{i=1}^{n_b}$ with their corresponding sizes $\{s_i\}_{i=1}^{n_b}$.

**for** $i = 1$ *to* $n_b$ **do**
    $j \leftarrow i - 1$
    **while** $j >= 0$ **do**
        **if** $j = 0$ **then**
            $PA(x_i) \leftarrow r$ /* Set $r$ as the parent node of $x_i$ */
            **break**
        **else if** $s_j > s_i$ **then**
            $PA(x_i) \leftarrow x_j$ /* Set $x_j$ as the parent node of $x_i$ */
            **break**
        **end**
        $j \leftarrow j - 1$
    **end**
**end**

**Output:** a tree $\mathcal{T}$ with root node $r$.

---

## B  Assumption Violation Examples

For Assumption 1, upon manually inspecting some samples, we found that errors in these particular cases were linked to the errors in the XY-cut algorithm (Ha et al., 1995), resulting in an incorrect arrangement of text blocks.

Figure A1 presents two examples where Assumption 2 is not satisfied. In the first example, the term "The way we work" serve as the parent-heading of "Corporate governance" which is a sub-heading, but featuring a smaller font size. Despite the smaller font size of "The way we work", it is clearly delineated from the sub-headings below by two green horizontal lines. However, our method focuses solely on text, neglecting visual cues such as these lines. In the second example, "COMMUNITY OUTREACH" is a sub-heading under "SOCIAL RESPONSIBILITY", but is has a larger font size, as this page emphasises community achievements.

Figure A2 presents two instances where Assumption 3 is violated. In the first example, "indirect economic impacts" and "Transmission System Investments" are headings situated at the same hierarchical level but have distinct font sizes. This dissimilarity could potentially lead to confusion for human readers, questioning whether these two headings should be placed within the same hierarchical level. In the second example, "Sustainability Fund purchases" and "Spend by solution type" are also headings at the same level, with subtly different font sizes, 11 and 10, respectively. While this difference may go unnoticed by humans, it does impact the performance of our method.

## C  Case Study

Figure A3 and Figure A4 illustrate a favourable scenario and an unfavorable one for CMM within the context of ESGDoc. The favourable case in Figure A3 demonstrates the capability of our model to handle lengthy document, where it generates a high quality tree structure.

Conversely, Figure A4 represents a challenging scenario where our model encounters difficulties across multiple nodes. Figure A5 further elaborated on this issue, showcasing four example pages of the unfavorable case illustrated in Figure A4. On the top-left page, CMM incorrectly retain the non-heading "ABOUT BRANDYWINE". This is a challenging case as "ABOUT BRANDYWINE" is prominently displayed in a large font at the top-left corner of the page, making it difficult to identify as a non-heading. A similar situation occurs on the top-right page, where CMM incorrectly keeps the non-heading "ENVIRONMENTAL PROGRESS". In this instance, "ENVIRONMENTAL PROGRESS" is enlarged to emphasise the company's achievements.

For the bottom two pages in Figure A5, CMM might encounter confusion between headings with coloured lead-in sentences. In the bottom-left page, "MANAGING CLIMATE RISK" functions as a heading and follows a pattern similar to other lead-in sentences such as "GOVERNANCE" and "STRATEGY AND RISK MANAGEMENT". They typically begin with a large, colored sentence followed by a paragraph. The bottom-right page presents a similar challenge. "OUR TENANTS" and "VALUED PARTNERSHIPS" share a similar pattern, with the former being a heading, whereas the latter not. The determination of "MANAGING CLIMATE RISK" and "OUR EMPLOYEES" as headings is primarily based on their position and

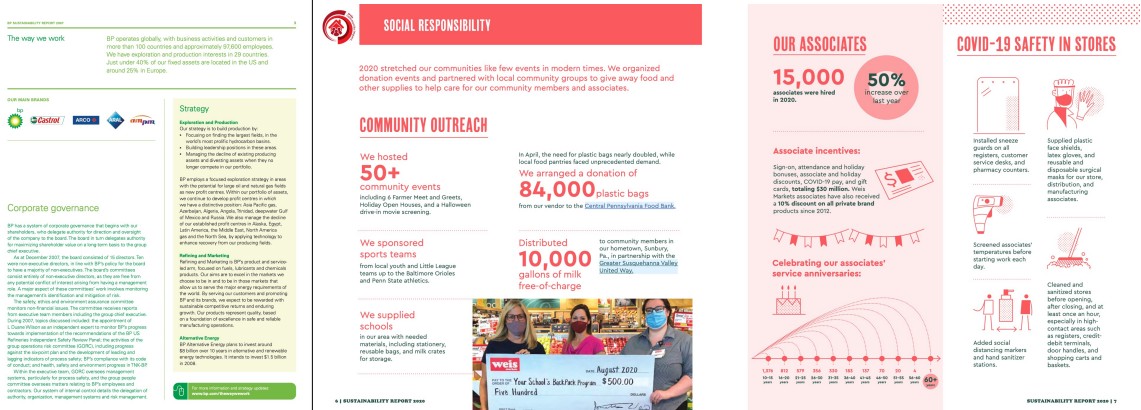

Figure A1: Two examples in `ESGDoc` where Assumption 2 is violated, one in portrait and the other in landscape orientation.

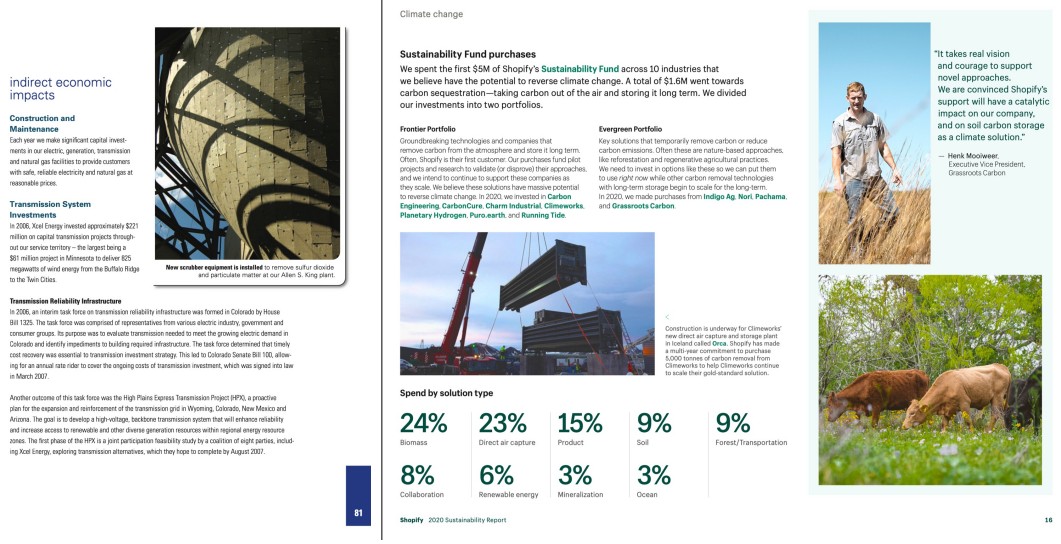

Figure A2: Two examples in `ESGDoc` where Assumption 3 is violated, one in portrait and the other in landscape orientation.

colour. However, it is worth noting that CMM does not use visual information, making it difficult for the model to handle such scenarios. Future work could explore the integration of visual information to enhance the model's performance in handling these situations.

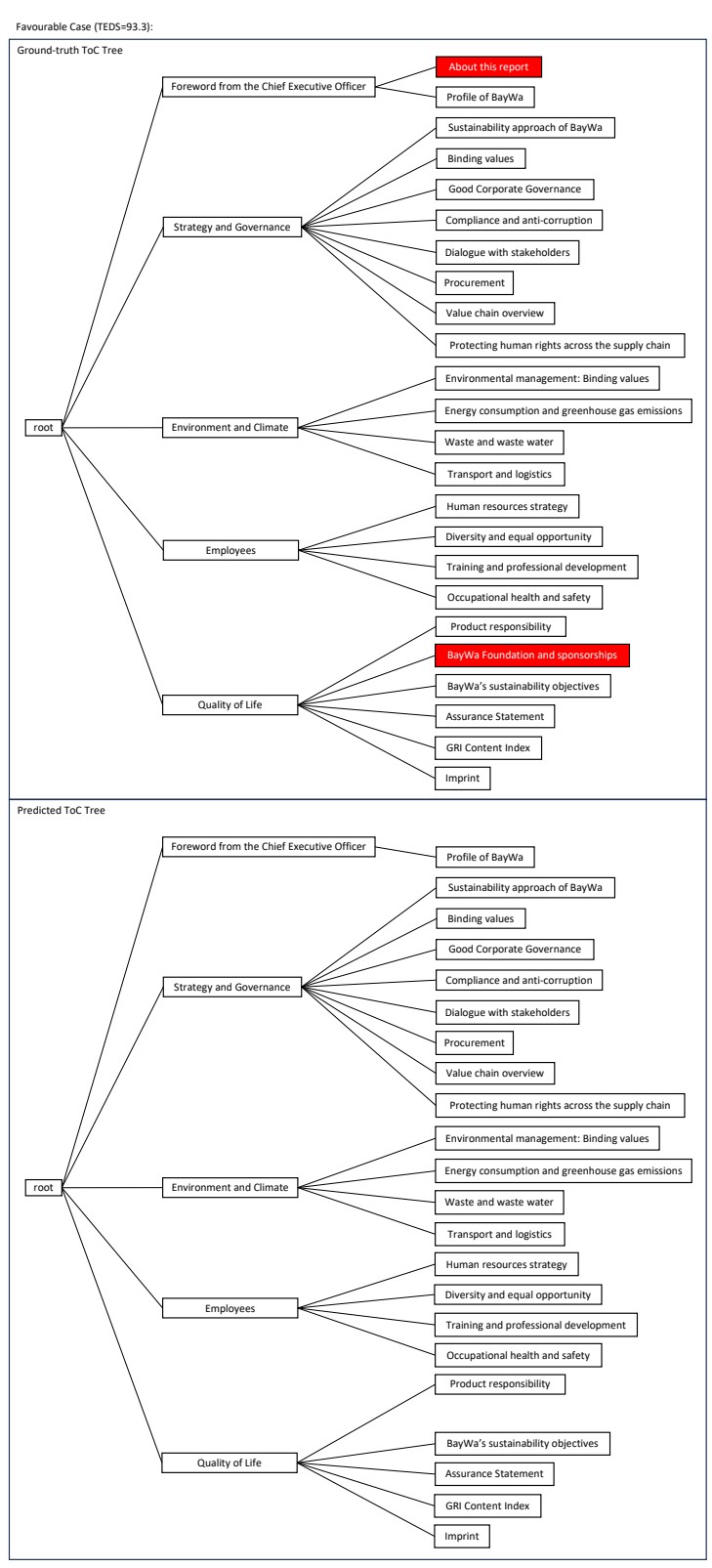

Figure A3: A favourable case for CMM on ESGDoc. Blocks highlighted in red represent incorrect predictions.

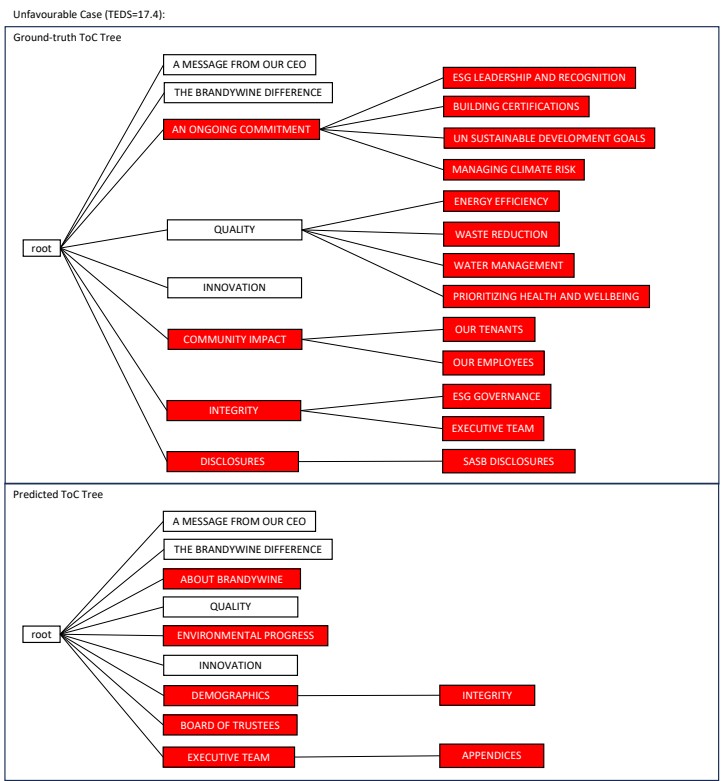

Figure A4: An unfavorable case for CMM on ESGDoc. Blocks highlighted in red represent incorrect predictions.

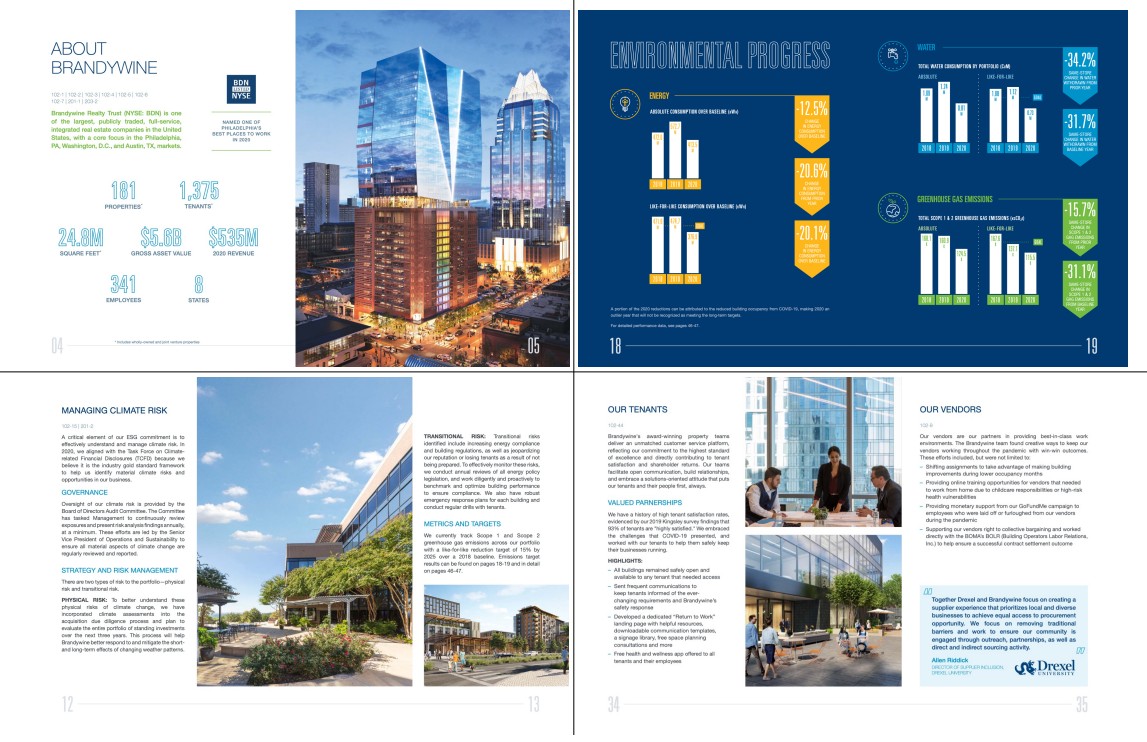

Figure A5: Four example pages of the unfavorable case highlighted in Figure A4. CMM preserves non-headings in the top two pages, while deleting headings in the bottom two pages.