# OpenReview forum: "A Scalable Framework for Table of Contents Extraction from Complex ESG Annual Reports"
_EMNLP/2023/Conference — EMNLP 2023 Main_

### Official Review · Reviewer_Eyfj · 2023-08-04

**Soundness:** 4

**Excitement:**

4: Strong: This paper deepens the understanding of some phenomenon or lowers the barriers to an existing research direction.

**Paper Topic And Main Contributions:**

The paper addresses the automatic generation of tables of contents
from PDF documents, extending previous work in the area, which has
focused heavily on scientific papers, with a corpus of corporate
annual reports (ESGDoc).  Compared with scientific documents,
documents in ESGDoc tend to be much longer and exhibit greater
variation in structure and presentational features.  Arguing that SOTA
approaches are too inefficient to apply to long documents, the paper
proposes an approach that involves applying a GNN to subgraphs that
represent the local neighborhood of particular document elements
(typically candidate headers suitable for inclusion in a TOC).  The
approach decides what to do with each such element individually and
constructs the resulting TOC deterministically based on these
decisions.  Experiments on two problem sets, a previously proposed
collection of scientific documents and the corpus of the paper, show
that that the proposed approach outperforms a leading SOTA model, for
which some of the longer documents prove to be intractable.  The
proposed approach is shown to be superior in both efficiency and
accuracy.

**Questions For The Authors:**

A. Your articulation of assumptions is very helpful, but your Assumption
3 seems a little problematic, based on Table 1.  To what extent does
the data's violation of this assumption depress your performance
scores?

B. Your statement that MTD "only uses" GRUs to capture candidate header
context is slightly confusing.  Did you mean to say that MTD only
models local context, or does the use of GRUs somehow contribute to
the limitation of MTD that you're trying to articulate?  More
generally, I think you need to say more about MTD, since it's your
only point of comparison with prior work.

**Reasons To Accept:**

The paper enriches prior work on automated TOC creation by
contributing a new domain that is considerably more difficult than
previously used problem sets.  Where accuracies on HierDoc hover
around 90%, raising concerns of saturation, performance on ESGDoc is
much lower, suggesting that it may be an interesting resource for
research into document understanding.

The approach proposed in the paper (CMM) is apparently effective.  The
paper musters convincing evidence that CMM should be preferred to MTD,
the SOTA model used as baseline in the paper.  Improvements over MTD
in both efficiency and accuracy are consistent and, in the case of
ESGDoc, substantial.

The algorithm is intuitive and its explication solid.  It should be
straightforward for a knowledgeable reader to implement the approach
and, with the data contributed by the paper, replicate the results.

**Reasons To Reject:**

The motivation for the work is underdeveloped.  In the introduction,
the paper merely notes that this task is a focus of prior work, but it
was unclear to me what solving this problem enables.  Certainly, the
research is of academic interest, and perhaps one can imagine a
document visualizer or summarization system based on this work, but
what else might it lead to?  This question is all the more acute for
the ESG domain, where accuracy is low and subject matter varies
widely.  In the scientific domain, because of strong structural
presentations, one can imagine basing downstream automation off the
interpretation of a paper, but is this possible in the contributed
domain?  What could you create from this functionality, other than a
kind of automated document thumbnail generator?

The experiments are not particularly extensive, employing one baseline
and two problem sets.  The baseline is not described in sufficient
detail for the reader to understand the tradeoffs implied by the
comparison of CMM and MTD.  The experiments seem to imply that CMM is
simply to be preferred in all cases, but there's not enough here for a
reader to reach that conclusion confidently.

The paper doesn't provide much in the way of insights into the ESGDoc
task.  It's evident that the problem is much harder than HierDoc, but
why?  I found myself wondering whether it's simply underspecified and
somewhat arbitrary what TOC emerges from a typical document in this
domain.  More concretely, how would humans do in performing this task?
If humans can reliably reconstruct the given TOC, then I think we have
the makings of an interesting challenge.  The paper doesn't offer much
of an error analysis to provide insights into why, for example, CMMs
ability to identify ESG TOC entries hovers around 50%.  Diversity of
presentation conventions?  Document length?  Something else that might
be address with better understanding of section contents?  The paper
doesn't provide satisfactory answers to these questions.

**Reproducibility:**

4: Could mostly reproduce the results, but there may be some variation because of sample variance or minor variations in their interpretation of the protocol or method.

**Reviewer Confidence:**

4: Quite sure. I tried to check the important points carefully. It's unlikely, though conceivable, that I missed something that should affect my ratings.

**Typos Grammar Style And Presentation Improvements:**

You might want to say explicitly that you're releasing ESGDoc to
support further research, rather than just "introducing" the dataset.

You talk about document understanding a fair amount in the front
matter, but never define it.  A succinct statement concerning what you
think research into document understanding is about (and what uses it
might have) would help make the scope of your research more concrete.

---

> ### Author Rebuttal · Authors · 2023-08-28
>
> > **Q1**: The motivation for the work is underdeveloped. In the introduction, the paper merely notes that this task is a focus of prior work, but it was unclear to me what solving this problem enables. Certainly, the research is of academic interest, and perhaps one can imagine a document visualizer or summarization system based on this work, but what else might it lead to? This question is all the more acute for the ESG domain, where accuracy is low and subject matter varies widely. In the scientific domain, because of strong structural presentations, one can imagine basing downstream automation off the interpretation of a paper, but is this possible in the contributed domain? What could you create from this functionality, other than a kind of automated document thumbnail generator?
>
> **A1**: The motivation behind the work is rooted in the observation that while traditional summarisation offers a concise representation of a document’s content, a Table of Contents (ToC) presents a structured and hierarchical summary. This structural organisation in a ToC provides a comprehensive pathway for pinpointing specific information. For example, when seeking information about a company's carbon dioxide emissions, a ToC enables a systematic navigation through the information hierarchy, such as environment -> climate -> greenhouse gases -> carbon dioxide. In contrast, conventional summarisation might only provide a vague indication of such information, requiring sifting through the entire document for precise detail.
>
> While our primary focus in this paper centres on the ESG domain, the automated generation of a ToC holds relevance for lengthy documents across various domains. Furthermore, the construction and modelling components introduced in our methodology hold potential for future applications in other document-related tasks.
>
>
> ---
> > **Q2**: The experiments are not particularly extensive, employing one baseline and two problem sets. The baseline is not described in sufficient detail for the reader to understand the tradeoffs implied by the comparison of CMM and MTD. The experiments seem to imply that CMM is simply to be preferred in all cases, but there's not enough here for a reader to reach that conclusion confidently.
>
> **A2**: We will add more detail about MTD in the paper. Our choice to limit the baseline to MTD is attributed to MTD proposing this task, and the limited existing work on this particular task.
>
> CMM holds a comparative advantage over MTD for two main reasons:
>
> (1) *Incorporation of auxiliary information in the modelling step*. CMM leverages size and three assumptions as additional cues to construct an initial tree, already imparting some hierarchical context. In contrast, MTD starts the modelling process from scratch, without leveraging such external cues.
>
> (2) *Task simplification in the modification step*. We simplify a task of paired heading classification in MTD into singular heading classification, resulting in a reduction of decoding time complexity from O(n^2) to O(n). However, such simplification suffers from the issue of assumption violation (see Section 5.2, Section Limitations, and Appendix C). The modification step, only containing three operations, does not completely fix the problem of assumption violation, which in turn limits the performance improvement in the modification step. To illustrate,  even with a flawless modification step, the ToC scores are 95.4% on HierDoc and 84.4% on ESGDoc. Nonetheless, in this paper, our primary focus is on presenting a feasible solution for dealing with lengthy documents with complex visual layouts and our model exhibits superior performance compared to MTD.
>
>
> ---
> > **Q3**: The paper doesn't provide much in the way of insights into the ESGDoc task. It's evident that the problem is much harder than HierDoc, but why? I found myself wondering whether it's simply underspecified and somewhat arbitrary what TOC emerges from a typical document in this domain. More concretely, how would humans do in performing this task? If humans can reliably reconstruct the given TOC, then I think we have the makings of an interesting challenge. The paper doesn't offer much of an error analysis to provide insights into why, for example, CMMs ability to identify ESG TOC entries hovers around 50%. Diversity of presentation conventions? Document length? Something else that might be address with better understanding of section contents? The paper doesn't provide satisfactory answers to these questions.
>
> **A3**: HierDoc's relatively straightforward nature can be attributed to the structured formatting found in  scientific papers, where heading numbering such as "1. Introduction" facilitate easier classification. This consistent structuring and numbering mechanism offers clues that models can easily rely on. (see Line 053-057, Line 539-546). On the other hand, ESGDoc, in contrast, exhibits significant layout variability. The content structure, the wording of headings, and their hierarchical ordering can differ widely, depending on the reporting company's chosen narrative (see Fig 1, A1, A2, and A5 as examples).
>
> Due to the page limitation, case study and error analysis are shown in Appendix C and D. The ToC of each ESG report depends on the style of the company and the narrative they aim to present. For instance, the phrase "About this report" can manifest in multiple ways across reports. In some instances, it might stand as an independent heading, while in others, it might be incorporated within the body.  This diversity can be perplexing, not only for models but even for human readers. More details are discussed in Appendix C and D.
>
> The ESGDoc dataset encompasses reports varying from 4 pages to 521 pages, with an average of 72 pages, whereas HierDoc ranges from 2 pages to 298 pages, with an average of 19 pages. (See Section 3). On average, ESGDoc is 3.8 times longer than HierDoc. Further elaboration on this aspect will be incorporated in the revised version.
>
>
> ---
> > **Q4**: Your articulation of assumptions is very helpful, but your Assumption 3 seems a little problematic, based on Table 1. To what extent does the data's violation of this assumption depress your performance scores?
>
> **A4**: We did not conduct such experiments to avoid impacting the original dataset distribution.  Modifying or removing data that doesn't conform to assumptions could result in altered data distributions that deviate from real-world data. Nevertheless, we can offer some insights into this matter. Assumption violation mainly affects the modification phase, as the modification with only three operations cannot address all assumption violation problems.
>
> Comparing CMM with MTD, CMM outperforms MTD in heading detection (HD) by 13.6% but only by 4.2% in ToC, as shown in Table 2. HD assessment only evaluates the construction and modelling steps, while ToC evaluation encompasses all three steps. The limited efficacy of the modification step in addressing assumption violations leads to a low ToC score. Moreover, as mentioned in A2, assumption violation results in a performance limit for the modification step, with ToC score of 95.4% for HierDoc and 84.4% for ESGDoc.
>
>
> ---
> > **Q5**: Your statement that MTD "only uses" GRUs to capture candidate header context is slightly confusing. Did you mean to say that MTD only models local context, or does the use of GRUs somehow contribute to the limitation of MTD that you're trying to articulate? More generally, I think you need to say more about MTD, since it's your only point of comparison with prior work.
>
> **A5**: MTD initially uses BERT to derive text representations for various segments of a document, including both the headings and bodies. Then, GRU is employed to model these representations, forming a sequence of vectors representing headings and bodies. The phrase "only uses" emphasises that MTD relies solely on the GRU to capture contextual connections among candidate headings from scratch, without leveraging hierarchical cues as mentioned in A2. While GRUs are adept at capturing sequential dependencies, their efficiency diminishes when dealing with extremely long sequences, as encountered in documents that comprise thousands of textual units, leading to potential gaps in comprehending context.
>
>
> ---
> > **Q6**: You might want to say explicitly that you're releasing ESGDoc to support further research, rather than just "introducing" the dataset.
>
> **A6**: Yes, we will release them as mentioned in the footnote of Line 32 in the paper.
>
>
> ---
> > **Q7**: You talk about document understanding a fair amount in the front matter, but never define it. A succinct statement concerning what you think research into document understanding is about (and what uses it might have) would help make the scope of your research more concrete.
>
> **A7**: Document understanding refers to interpreting and comprehending the structure, content, and semantics of documents. Previous methods of document understanding might prioritise layout details, such as classifying titles, figures, tables, and body text. Conversely, in this paper, we focus more on semantic comprehension, which encompasses not just recognising individual words or sentences but also grasping the hierarchical relationships and context within the document. It's akin to how humans process lengthy reads: we don't just understand isolated facts, but also how those facts interrelate and the overarching narratives they collectively form.

---

### Official Review · Reviewer_eEPP · 2023-08-05

**Soundness:** 3

**Excitement:**

4: Strong: This paper deepens the understanding of some phenomenon or lowers the barriers to an existing research direction.

**Paper Topic And Main Contributions:**

The paper talks about the challenges faced by the existing system in extracting Table of Contents (ToC) from long and complex PDF documents and offers a new scalable ToC extraction method. The proposed approach consists of constructing a tree structure of the extracted blocks of text from the PDF, parsing through sub-trees and capturing each node's contextual information and finally structuring a new tree with the original nodes being either kept as is, deleted or moved. This work is supplemented by introducing a new dataset called ESGDoc that consists of 1093 ESG (Environmental, Social, and Governance) annual reports from 563 companies. The paper shows how their approach improves over the baseline on the ESGDoc as well as on the existing HierDoc datasets in terms of both accuracy as well as the ability to handle longer length documents.

**Questions For The Authors:**

Question A: Have the authors checked if instead of decomposing the PDF into blocks and then constructing the tree, instead convert the PDF into say HTML and use the DOM tree? Would this avoid the initial tree construction in Algorithm 2?

Question B: Figure A3 shows ground truth ToC tree of ESGDoc. What are the criteria used for annotating {Keep, Move, Delete} other than those described in \subsection {Tree Modification}?

Question C: Do you have insights on why the TEDS performance jumps in Figure 3 when d goes from 1 to 2 for n_d and shows only a minute increase afterwards?




**Reasons To Accept:**

The authors have introduced the idea of splitting the original PDF tree into sub-trees that will allow to parse large PDFs without facing out of memory errors.

The authors have modified their custom dataset that would otherwise have not been able to handle by the baseline approach during evaluation.

Contribution towards a new dataset that focuses on a more complex and longer document size than the existing one.

A significant improvement in training time and performance over the baseline.


**Reasons To Reject:**

None.

**Reproducibility:**

3: Could reproduce the results with some difficulty. The settings of parameters are underspecified or subjectively determined; the training/evaluation data are not widely available.

**Reviewer Confidence:**

3: Pretty sure, but there's a chance I missed something. Although I have a good feel for this area in general, I did not carefully check the paper's details, e.g., the math, experimental design, or novelty.

**Typos Grammar Style And Presentation Improvements:**

In line 615 instead of n_{d} > 3 it can be n_{d} \geq 3 due to only a minute improvement witnessed in Figure 3.

It would be helpful to add the inference time as well in addition to the existing training time since in the paper run-time is the term used instead of training time.

---

> ### Author Rebuttal · Authors · 2023-08-28
>
> > **Q1**: Have the authors checked if instead of decomposing the PDF into blocks and then constructing the tree, instead convert the PDF into say HTML and use the DOM tree? Would this avoid the initial tree construction in Algorithm 2?
>
> **A1**: The distinction between the DOM tree and our tree construction methodology holds significant importance. The DOM tree primarily focuses on the layout details and the storage of the PDF file. On the other hand, our constructed tree prioritises semantic content organisation, aligning with human cognitive patterns.
>
> Let’s consider an example: Heading 2., as a first-level heading, shares a stronger semantic relation with other first-level headings, such as Heading 5. However, its relation to a second-level heading, say 4.2, is weaker, despite the possibility of Heading 4.2 being physically closer to Heading 2 in terms of page sequence. The extent of such correlations can be adjusted by modifying the parameter n_d. Unfortunately, the DOM tree falls short in capturing these nuanced semantic relationships.
>
>
> ---
> > **Q2**: Figure A3 shows ground truth ToC tree of ESGDoc. What are the criteria used for annotating {Keep, Move, Delete} other than those described in \subsection {Tree Modification}?
>
> **A2**: To begin, we annotate all headings as “keep” and non-headings as “delete”. Then for each heading, we check if its ground-truth parent heading corresponds to one of its preceding siblings in the original tree. If yes, we change its annotation from “keep” to “move”.
>
>
> ---
> > **Q3**: Do you have insights on why the TEDS performance jumps in Figure 3 when d goes from 1 to 2 for n_d and shows only a minute increase afterwards?
>
> **A3**: The primary factors contributing to this phenomenon may include:
> (1) A significant portion of documents exhibit a linear structure. To illustrate,  when n_d=2, it corresponds to a hierarchical arrangement featuring primary headings, secondary headings, and main body  content in a three-tier configuration.
> (2) The constructed initial tree inherently positions related headings in close proximity according to their semantic relationships, without regard for their relative page placement. As a consequence, a heading's most relevant contextual information predominantly emerges from its immediate neighbours within the tree.
>
> For example, when examining heading 1.2., information from heading 1. (n_d=1) offers a comprehensive overview of the encompassing chapter. Simultaneously, heading 2. (n_d=2) can provide supplementary insights, such as affirming that heading 1.2. is nested within the domain of heading 1., rather than heading 2. However, delving into deeper levels may become redundant. For example, a heading like 2.2. (n_d=3), situated more distantly in the semantic space, would not notably  enhance the understanding of heading 1.1.
>
>
> ---
> > **Q4**: It would be helpful to add the inference time as well in addition to the existing training time since in the paper run-time is the term used instead of training time.
>
> **A4**: The inference time in terms of the number of minutes is as follows:
> |  | HierDoc | ESGDoc P. |
> |---|---|---|
> | MTD | 16.1 |  2.7 |
> | CMM | 3.8 | 2.1 |
> | Ratio | 4.2x | 1.3x |
> We will add it to the revised version.

---

### Official Review · Reviewer_Z2mu · 2023-08-05

**Soundness:** 4

**Excitement:**

4: Strong: This paper deepens the understanding of some phenomenon or lowers the barriers to an existing research direction.

**Missing References:**

-

**Paper Topic And Main Contributions:**

This paper presents a scalable framework for Table of Contents (ToC) extraction from ESG reports. The main contributions include the introduction of a new dataset, ESGDoc, a novel framework for processing documents in a construction-modelling-modification manner, and a graph-based method for document segmentation and modelling.

**Questions For The Authors:**

Have you considered releasing the codebase and dataset for your approach?

**Reasons To Accept:**

The paper demonstrates a strong baseline dataset comparison, an ablation study and a solid methodology foundation to address the problem effectively. The experimental design and results are presented clearly and concisely. Additionally, the detailed elaboration on the dataset construction enhances the reproducibility of the research.

**Reasons To Reject:**

A comparison with other state-of-the-art models would provide valuable insights and further strengthen the contribution of the proposed framework.

**Reproducibility:**

4: Could mostly reproduce the results, but there may be some variation because of sample variance or minor variations in their interpretation of the protocol or method.

**Reviewer Confidence:**

3: Pretty sure, but there's a chance I missed something. Although I have a good feel for this area in general, I did not carefully check the paper's details, e.g., the math, experimental design, or novelty.

**Typos Grammar Style And Presentation Improvements:**

Please ensure consistent use of abbreviations throughout the paper. For example, use "ToC" (001) instead of "Toc"(002) for Table of Contents (ToC) in all sections of the manuscript.

---

> ### Author Rebuttal · Authors · 2023-08-28
>
> > **Q1**: A comparison with other state-of-the-art models would provide valuable insights and further strengthen the contribution of the proposed framework.
>
> **A1**: The task we've undertaken is relatively novel and the only existing approach applied on the task that we are aware of is MTD. Although we did explore other potential baselines such as chatPDF as well as certain specialised plugins of chatGPT and GPT4, these methodologies have not been made available to the public. Furthermore, both chatGPT and GPT4 inherently come with a constraint on input  length, which prevents them from effectively processing extensive documents all at once. Due to these limitations, we made the decision to not consider them as baselines in our study.
>
> ---
> > **Q2**: Have you considered releasing the codebase and dataset for your approach?
>
> **A2**: Yes, we will release them as mentioned in the footnote of Line 32.

---

### Meta-Review · Area_Chair_Se2V · 2023-09-06

**Recommendation:** 4

**Metareview:**

This paper presents a strong baseline dataset comparison, includes an ablation study, has a solid methodology foundation, and provides clear and concise experimental design and results. However, this paper has limitations in a lack of comparison with other state-of-the-art models, which would provide valuable insights and further strengthen the contribution of the proposed framework.

---

### Decision · Program_Chairs · 2023-10-07

**Decision:**

Accept-Main

**Comment:**

This paper presents a strong baseline dataset comparison, includes an ablation study, has a solid methodology foundation, and provides clear and concise experimental design and results. However, this paper has limitations in a lack of comparison with other state-of-the-art models, which would provide valuable insights and further strengthen the contribution of the proposed framework.